# The Limits of Post-Selection Generalization

**Kobbi Nissim**[*]
Georgetown University
kobbi.nissim@georgetown.edu

**Adam Smith**[†]
Boston University
ads22@bu.edu

**Thomas Steinke**
IBM Research – Almaden
posel@thomas-steinke.net

**Uri Stemmer**[‡]
Ben-Gurion University
u@uri.co.il

**Jonathan Ullman**[§]
Northeastern University
jullman@ccs.neu.edu

## Abstract

While statistics and machine learning offers numerous methods for ensuring generalization, these methods often fail in the presence of *post selection*—the common practice in which the choice of analysis depends on previous interactions with the same dataset. A recent line of work has introduced powerful, general purpose algorithms that ensure a property called *post hoc generalization* (Cummings *et al.*, COLT'16), which says that no person when given the output of the algorithm should be able to find any statistic for which the data differs significantly from the population it came from.

In this work we show several limitations on the power of algorithms satisfying post hoc generalization. First, we show a tight lower bound on the error of any algorithm that satisfies post hoc generalization and answers adaptively chosen statistical queries, showing a strong barrier to progress in post selection data analysis. Second, we show that post hoc generalization is not closed under composition, despite many examples of such algorithms exhibiting strong composition properties.

## 1 Introduction

Consider a dataset $X$ consisting of $n$ independent samples from some unknown population $\mathcal{P}$. How can we ensure that the conclusions drawn from $X$ *generalize* to the population $\mathcal{P}$? Despite decades of research in statistics and machine learning on methods for ensuring generalization, there is an increased recognition that many scientific findings do not generalize, with some even declaring this to be a "statistical crisis in science" [14]. While there are many reasons a conclusion might fail to generalize, one that is receiving increasing attention is *post-selection*, in which the choice of method for analyzing the dataset depends on previous interactions with the same dataset. Post-selection can arise from many common practices, such as *variable selection*, *exploratory data analysis*, and *dataset re-use*. Unfortunately, post-selection invalidates traditional methods for ensuring generalization, which assume that the method is independent of the data.

Numerous methods have been devised for statistical inference after post selection (e.g. [16, 18, 12, 13, 23]). These are primarily *special purpose* procedures that apply to specific types of simple post selection that admit direct analysis. A more limited number of methods apply where the data is reused in one of a small number of prescribed ways (e.g. [2, 4]).

[*]Supported by NSF award CNS-1565387.

[†]Supported by NSF awards IIS-1447700 and AF-1763665, a Google Faculty Award and a Sloan Foundation Research Award.

[‡]Work done while U.S. was a postdoctoral researcher at the Weizmann Institute of Science, supported by a Koshland fellowship, and by the Israel Science Foundation (grants 950/16 and 5219/17).

[§]Supported by NSF awards CCF-1718088, CCF-1750640, and CNS-1816028, and a Google Faculty Award.

A recent line of work initiated by Dwork *et al.* [9] posed the question: Can we design *general-purpose* algorithms for ensuring generalization in the presence of post-selection? These works (e.g. [9, 8, 19, 1]) identified properties of an algorithm that ensure generalization under post-selection, including *differential privacy* [10], information-theoretic measures, and compression. They also identified many powerful general-purpose algorithms satisfying these properties, leading to algorithms for post-selection data analysis with greater statistical power than all previously known approaches.

Each of the aforementioned properties give incomparable generalization guarantees, and allow for qualitatively different types of algorithms. However, Cummings *et al.* [7] identified that the common thread in each of these approaches is to establish a notion of *post hoc generalization* (which they originally called *robust generalization*), and initiated a general study of algorithms satisfying this notion. Informally, an algorithm $\mathcal{M}$ satisfies post hoc generalization if there is no way, given only the output of $\mathcal{M}(X)$, to identify any *statistical query* [17] (that is, a bounded, linear, real-valued statistic on the population) such that the value of that query on the dataset is significantly different from its answer on the whole population.

**Definition 1.1** (Post Hoc Generalization [7]). *An algorithm $\mathcal{M} : \mathcal{X}^n \to \mathcal{Y}$ satisfies $(\varepsilon, \delta)$-post hoc generalization if for every distribution $\mathcal{P}$ over $\mathcal{X}$ and every algorithm $\mathcal{A}$ that outputs a bounded function $q : \mathcal{X} \to [-1, 1]$, if $X \sim \mathcal{P}^{\otimes n}, y \sim \mathcal{M}(X)$, and $q \sim \mathcal{A}(y)$, then $\mathbb{P}\left[|q(\mathcal{P}) - q(X)| > \varepsilon\right] \leq \delta$, where we use the notation $q(\mathcal{P}) = \mathbb{E}\left[q(X)\right]$ and $q(X) = \frac{1}{n} \sum_i q(X_i)$, and the probability is over the sampling of $X$ and any randomness of $\mathcal{M}, \mathcal{A}$.*

Post hoc generalization is easily satisfied whenever $n$ is large enough to ensure *uniform convergence* for the class of statistical queries. However, uniform convergence is only satisfied in the unrealistic regime where $n$ is much larger than $|\mathcal{X}|$. Algorithms that satisfy post hoc generalization are interesting in the realistic regime where there will *exist* queries $q$ for which $q(\mathcal{P})$ and $q(X)$ are far, but these queries cannot be *found*. The definition also extends seamlessly to richer types of statistics than statistical queries. However, restricting to statistical queries only strengthens our negative results.

Since all existing general-purpose algorithms for post-selection data analysis are analyzed via post hoc generalization, it is crucial to understand what we can achieve with algorithms satisfying post hoc generalization. In this work we present several strong limitaitons on the power of such algorithms. Our results identify natural barriers to progress in this area, and highlight important challenges for future research on post-selection data analysis.

## 1.1 Our Results

**Sample Complexity Bounds for Statistical Queries.** Our first contribution is strong new lower bounds on any algorithm that satisfies post hoc generalization and answers a sequence of adaptively chosen statistical queries—the setting introduced in Dwork *et al.* [9] and further studied in [1, 15, 20]. In this model, there is an underlying distribution $\mathcal{P}$. We would like to design an algorithm $\mathcal{M}$ that holds a sample $X \sim \mathcal{P}^{\otimes n}$, takes statistical queries $q$, and returns accurate answers $a$ such that $a \approx q(\mathcal{P})$. To model post-selection, we consider a *data analyst* $\mathcal{A}$ that issues a sequence of queries $q^1, \ldots, q^k$ where each query $q^j$ may depend on the answers $a^1, \ldots, a^{j-1}$ given by the algorithm in response to previous queries.

The simplest algorithm $\mathcal{M}$ for this task of answering adaptive statistical queries would return the empirical mean $q^j(X) = \frac{1}{n} \sum_i q^j(X_i)$ in response to each query, and one can show that this algorithm answers each query to within $\pm\varepsilon$ if $n \geq \tilde{O}(k/\varepsilon^2)$ samples. Surprisingly, we can improve the sample complexity to $n \geq \tilde{O}(\sqrt{k}/\varepsilon^2)$ by returning $q(X)$ perturbed with carefully calibrated noise [9, 1]. The analysis of this approach uses post hoc generalization: the noise is chosen so that $|a - q(X)| \leq \varepsilon/2$ and the noise ensures $|q(\mathcal{P}) - q(X)| \leq \varepsilon/2$ for every query the analyst asks.

Our main result shows that the sample complexity $n = \tilde{O}(\sqrt{k}/\varepsilon^2)$ is essentially optimal for *any* algorithm that uses the framework of post hoc generalization.

**Theorem 1.2** (Informal). *If $\mathcal{M}$ takes a sample of size $n$, satisfies $(\varepsilon, \delta)$-post hoc generalization, and for every distribution $\mathcal{P}$ over $\mathcal{X} = \{\pm 1\}^{k+O(\log(n/\varepsilon))}$ and every data analyst $\mathcal{A}$ who asks $k$ statistical queries, $\mathbb{P}\left[\exists j \in [k], |q^j(\mathcal{P}) - a| > \varepsilon\right] \leq \delta$ then $n = \Omega(\sqrt{k}/\varepsilon^2)$, where the probability is taken over $X \sim \mathcal{P}^{\otimes n}$ and the coins of $\mathcal{M}$ and $\mathcal{A}$.*

To prove our theorem, we construct a joint distribution over pairs $(\mathcal{A}, \mathcal{P})$ such that when $\mathcal{M}$ is given too small a sample $X$, and $\mathcal{A}$ asks $k - 1$ statistical queries, then either $\mathcal{M}$ does not answer all the

queries accurately or $\mathcal{A}$ outputs a $k$-th query $q^*$ such that $q^*(\mathcal{P}) - q^*(X) > \varepsilon$. Thus, $\mathcal{M}$ cannot be both accurate and satisfy post hoc generalization.

Our proof of this result refines the techniques in [15, 20]—which yield a lower bound of $n = \Omega(\sqrt{k})$ for $\varepsilon = 1/3$.

Our proof circumvents a barrier in previous lower bounds. The previous works use the sequence of queries to uncover almost all of the sample held by the mechanism (a "reconstruction attack" of sorts). Once the analyst has identified all the points in the sample, it is easy to force an error: the analyst randomly asks one of two queries – zero everywhere or zero on the reconstructed sample and one elsewhere – that "look the same to" $\mathcal{M}$ but have different true answers.

We cannot use that approach because in our setting it is *impossible* to reconstruct *any* of the sample. Indeed, for the parameter regime we consider, differentially private algorithms could be used to prevent reconstruction with any meaningful confidence. All we can hope for is a weak *approximate* reconstruction of the sample. This means the algorithm will have sufficient information to distinguish the aforementioned two queries and we cannot end the proof the same way as previously.

Intuitively, our attack approximately reconstructs the dataset in a way that is only $O(\varepsilon)$ better than guessing. This is not enough to completely "cut off" the algorithm and force an error, but, as we will see, does allow the analyst to construct a query $q^*$ that overfits – i.e., $|q^*(X) - q^*(P)| > \varepsilon$. Our approximate reconstruction is accomplished using a modification of the reconstruction attack techniques of prior work. Specifically, we employ tools from the fingerprinting codes literature [3, 22, 6] but we output quantitative scores, rather than a hard in/out decision about what is in the sample.

Independently, Wang [24] proved a quantitatively similar bound to Theorem 1.2. However, Wang's bound only applies to algorithms $\mathcal{M}$ that receive only the empirical mean $q(X)$ of each query (as opposed to the whole data set). This precludes mechanisms such as sample splitting that treat records assymetrically. Wang's bound also applies for a slightly different (though closely related) class of statistics.

The dimensionality of $\mathcal{X}$ required in our result is at least as large as $k$, which is somewhat necessary. Indeed, if the support of the distribution is $\{\pm 1\}^d$, then there is an algorithm $\mathcal{M}$ that takes a sample of size just $\tilde{O}(\sqrt{d}\log(k)/\varepsilon^3)$ [9, 1], so the conclusion is simply false if $d \ll k$. Even when $d \ll k$, the aforementioned algorithms require running time at least $2^d$ per query. [15, 20] also showed that any *polynomial time* algorithm that answers $k$ queries to constant error requires $n = \Omega(\sqrt{k})$. We improve this result to have the optimal dependence on $\varepsilon$.

**Theorem 1.3** (Informal). *Assume one-way functions exist and let $c > 0$ be any constant. If $\mathcal{M}$ takes a sample of size $n$, has polynomial running time, satisfies $(\varepsilon, \delta)$-post hoc generalization, and for every distribution $\mathcal{P}$ over $\mathcal{X} = \{\pm 1\}^{k^c + O(\log(n/\varepsilon))}$ and every data analyst $\mathcal{A}$ who asks $k$ statistical queries, $\mathbb{P}\left[\exists j \in [k], |q^j(\mathcal{P}) - a| > \varepsilon\right] \leq \delta$, then $n = \Omega(\sqrt{k}/\varepsilon^2)$, where the probability is taken over $X \sim \mathcal{P}^{\otimes n}$ and the coins of $\mathcal{M}$ and $\mathcal{A}$.*

We prove the information-theoretic result (Theorem 1.2) in Section 2. Due to space restrictions, we defer the computational result (Theorem 1.3) to the full version of this work.

**Negative Results for Composition.**   Differential privacy provides optimal or near-optimal methods for answering an adaptively-chosen sequence of statistical queries. However, even for answering statistical queries, outside constraints sometimes preclude randomized algorithms (to allay reproducibility concerns, for instance). Furthermore, one of the main goals of the emerging study of adaptive data analysis is to understand unstructured, unplanned dataset re-use.

At this point, we know several techniques for reasoning about generalization in the adaptive setting: differential privacy and algorithmic stability, information bounds, and compression (and there may be many more yet to be discovered) [7]. These techniques are not directly comparable, but they all use posthoc generalization as a fundamental unit of their analysis. If posthoc generalization were to compose well, then this would provide an avenue for combining these techniques (and possibly others). However, we show that this is not the case and, hence, we must search elsewhere for a unifying theory.

Intuitively, we show that, if the same dataset is analyzed by many different algorithms each satisfying post hoc generalization, then the *composition* of these algorithms may not satisfy post hoc generaliza-

tion. That is, combining the information output by several algorithms may permit overfitting even when the individual outputs do not.

The key reason differential privacy is used for adaptive data analysis is that it satisfies strong composition properties – this is what quantitatively distinguishes the technique from, say, data splitting. We show that posthoc generalization does not have even weak adaptive composition properties. This shows a stark difference between differential privacy and posthoc generalization as tools for analyzing adaptive data analysis. This result can be viewed as further motivation for using differential privacy in this setting – its composition properties are special.

Theorem 1.4 states that there is a set of $O(\log n)$ algorithms that have almost optimal post hoc generalization, but whose composition does not have any non-trivial post hoc generalization.

**Theorem 1.4.** *For every $n \in \mathbb{N}$ there is a collection of $\ell = O(\log n)$ algorithms $\mathcal{M}_1, \ldots, \mathcal{M}_\ell$ that take $n$ samples from a distribution over $\mathcal{X} = \{0,1\}^{O(\log n)}$ such that (1) each of these algorithms are $(\varepsilon, \delta)$-post hoc generalizing for every $\delta > 0$ and $\varepsilon = O(\sqrt{\log(n/\delta)/n^{.999}})$, but (2) the composition $(\mathcal{M}_1, \ldots, \mathcal{M}_\ell)$ is not $(1.999, .999)$-post hoc generalizing.*

If we consider a relaxed notion of *computational post hoc generalization*, then we show that composition can fail even for just two algorithms. Informally, computational post hoc generalization means that Definition 1.1 is satisfied when the algorithm $\mathcal{A}$ runs in polynomial time.

**Theorem 1.5.** *Assume one-way functions exist. For every $n \in \mathbb{N}$ there are two algorithms $\mathcal{M}_1, \mathcal{M}_2$ that take $n$ samples from a distribution over $\mathcal{X} = \{0,1\}^{O(\log n)}$ such that (1) both algorithms are $(\varepsilon, \delta)$-computationally post hoc generalizing for every $\delta > n^{-O(1)}$ and $\varepsilon = O(\sqrt{\log(n/\delta)/n^{.999}})$, but (2) the composition $(\mathcal{M}_1, \mathcal{M}_2)$ is not $(1.999, .999)$-computationally post hoc generalizing.*

We prove the information-theoretic result (Theorem 1.4) in Section 3. Due to space restrictions, we defer the computational result (Theorem 1.5) to the full version of this work.

## 2 Lower Bounds for Statistical Queries

### 2.1 Post Hoc Generalization for Adaptive Statistical Queries

We are interested in the ability of *interactive* algorithms satisfying post hoc generalization to answer a sequence of statistical queries. Definition 1.1 applies to such algorithms via the following experiment.

---
**Algorithm 1:** $\mathsf{AQ}_{\mathcal{X},n,k}[\mathcal{M} \leftrightharpoons \mathcal{A}]$

---
$\mathcal{A}$ chooses a distribution $\mathcal{P}$ over $\mathcal{X}$
$X \sim \mathcal{P}^{\otimes n}$ and $X$ is given to $\mathcal{M}$ (but not to $\mathcal{A}$)
**For** $j = 1, \ldots, k$
    $\mathcal{A}$ outputs a statistical query $q^j$ (possibly depending on $q^1, a^1, \ldots, q^{j-1}, a^{j-1}$)
    $\mathcal{M}(X)$ outputs $a^j$

---

**Definition 2.1.** An algorithm $\mathcal{M}$ is $(\varepsilon, \delta)$-*post hoc generalizing for $k$ adaptive queries over $\mathcal{X}$ given $n$ samples* if for every adversary $\mathcal{A}$, $\displaystyle \mathop{\mathbb{P}}_{\mathsf{AQ}_{\mathcal{X},n,k}[\mathcal{M} \leftrightharpoons \mathcal{A}]} \left[ \exists j \in [k] \ \left| q^j(X) - q^j(\mathcal{P}) \right| > \varepsilon \right] \leq \delta$.

### 2.2 A Lower Bound for Natural Algorithms

We begin with an information-theoretic lower bound for a class of algorithms $\mathcal{M}$ that we call *natural algorithms*. These are algorithms that can only evaluate the query on the sample points they are given. That is, an algorithm $\mathcal{M}$ is *natural* if, when given a sample $X = (X_1, \ldots, X_n)$ and a statistical query $q : \mathcal{X} \to [-1, 1]$, the algorithm $\mathcal{M}$ returns an answer $a$ that is a function only of $(q(X_1), \ldots, q(X_n))$. In particular, it cannot evaluate $q$ on data points of its choice. Many algorithms in the literature have this property. Formally, we define natural algorithms via the game $\mathsf{NAQ}_{\mathcal{X},n,k}[\mathcal{M} \leftrightharpoons \mathcal{A}]$. This game is identical to $\mathsf{AQ}_{\mathcal{X},n,k}[\mathcal{M} \leftrightharpoons \mathcal{A}]$ except that when $\mathcal{A}$ outputs $q^j$, $\mathcal{M}$ does not receive all of $q^j$, but instead receives only $q_X^j = (q^j(X_1), \ldots, q^j(X_n))$.

**Theorem 2.2** (Lower Bound for Natural Algorithms). *There is an adversary $\mathcal{A}_{\mathsf{NAQ}}$ such that for every natural algorithm $\mathcal{M}$, and for universe size $N = 8n/\varepsilon$, if*

$$\mathbb{P}_{\mathsf{NAQ}_{[N],n,k}[\mathcal{M} \leftrightarrows \mathcal{A}_{\mathsf{NAQ}}]} \left[ \exists j \in [k] \ \ \left| q^j(X) - q^j(\mathcal{P}) \right| > \varepsilon \bigvee \left| a^j - q^j(\mathcal{P}) \right| > \varepsilon \right] \leq \tfrac{1}{100}$$

*then $n = \Omega(\sqrt{k}/\varepsilon^2)$. Here the sample $X$ is chosen via the game $\mathsf{NAQ}_{[N],n,k}$ (it is sampled uniformly from the domain $[N]$).*

The proof uses the analyst $\mathcal{A}_{\mathsf{NAQ}}$ described in Algorithm 2. For notational convenience, $\mathcal{A}_{\mathsf{NAQ}}$ actually asks $k + 1$ queries, but this does not affect the final result.

---

**Algorithm 2:** $\mathcal{A}_{\mathsf{NAQ}}$

---

**Parameters:** sample size $n$, universe size $N = \frac{8n}{\varepsilon}$, number of queries $k$, target accuracy $\varepsilon$

Let $\mathcal{P} \leftarrow \mathsf{U}_{[N]}$, $A^1 \leftarrow \emptyset$, and $\tau \leftarrow 9\varepsilon\sqrt{2k\log(\frac{96}{\varepsilon})} + 1$
**For** $j \in [k]$
  Sample $p^j \sim \mathsf{U}_{[0,1]}$
  **For** $i \in [N]$
    Sample $\tilde{q}_i^j \sim \mathsf{Ber}(p^j)$ and let $q^j(i) \leftarrow \left\{ \begin{array}{ll} \tilde{q}_i^j & i \notin A^j \\ 0 & i \in A^j \end{array} \right.$
  Ask query $q^j$ and receive answer $a^j$
  **For** $i \in [N]$
    Let $z_i^j \leftarrow \left\{ \begin{array}{ll} trunc_{3\varepsilon}(a^j - p^j) \cdot (q_i^j - p^j) & i \notin A^j \\ 0 & i \in A^j \end{array} \right.$
      where $trunc_{3\varepsilon}(x)$ takes $x \in \mathbb{R}$ and returns the nearest point in $[-3\varepsilon, 3\varepsilon]$ to $x$.
    Let $A^{j+1} \leftarrow \left\{ i \in [N] : \left| \sum_{\ell=1}^{j} z_i^\ell \right| > \tau - 1 \right\}$ (*N.B.* By construction, $A^j \subseteq A^{j+1}$.)

**For** $i \in [N]$
  Define $z_i \leftarrow \sum_{j=1}^{k} z_i^j$ and $q_i^* \leftarrow \frac{z_i}{\tau}$
Let $q^* : [N] \to [-1, 1]$ be defined by $q^*(i) \leftarrow q_i^*$

---

In order to prove Theorem 2.2, it suffices to prove that either the answer $a^j$ to one of the initial queries $q^j$ fails to be accurate (in which case $\mathcal{M}$ is not accurate, or that the final query $q^*$ gives significantly different answers on $X$ and $\mathcal{P}$ (in which case $\mathcal{M}$ is not robustly generalizing). Formally, we have the following proposition.

**Proposition 2.3.** *For an appropriate choice of $k = \Theta(\varepsilon^4 n^2)$ and $n, \frac{1}{\varepsilon}$ sufficiently large, for any natural $\mathcal{M}$, with probability at least $2/3$, either (1) $\exists j \in [k] \ \ |a^j - q^j(\mathcal{P})| > \varepsilon$, or (2) $q^*(X) - q^*(\mathcal{P}) > \varepsilon$. where the probability is taken over the game $\mathsf{NAQ}_{X,n,k}[\mathcal{M} \leftrightarrows \mathcal{A}_{\mathsf{NAQ}}]$ and $\mathcal{A}_{\mathsf{NAQ}}$ is specified by Algorithm 2.*

We prove Proposition 2.3 using a series of claims. The first claim states that none of the values $z_i$ are ever too large in absolute value, which follows immediately from the definition of the set $A^j$ and the fact that each term $z_i^j$ is bounded.

**Claim 2.4.** *For every $i \in [N]$, $|z_i| \leq \tau$.*

The next claim states that, no matter how the mechanism answers, very few of the items not in the sample get "accused" of membership, that is, included in the set $A^j$.

**Claim 2.5** (Few Accusations). $\Pr(|A_k \setminus X| \leq \varepsilon N/8) \geq 1 - e^{-\Omega(\varepsilon n)}$.

*Proof.* Fix the biases $p^1, ..., p^k$ as well as the all the information visibile to the mechanism (the query values $\{q_i^j : i \in X, j \in [k]\}$, as well as the answers $a^1, ..., a^k$). We prove that the probability of $F$ is high conditioned on any setting of these variables.

The main observation is that, once we condition on the biases $p^j$, the query values at $\{q_i^j : i \notin X, j \in [k]\}$ are independent with $q_i^j \sim \mathsf{Ber}(p^j)$. This is true because $\mathcal{M}$ is a natural algorithm (so it sees

| | |
|---|---|
| $n$ | Sample size. |
| $N$ | Universe size. |
| $\mathcal{P}$ | Target distribution, uniform on $[N]$. |
| $A_j$ | Universe elements "suspected" of being in the sample during the $j$th round of the attack. |
| $q^j$ | The query constructed in the $j$ round. |
| $z_i^j$ | If $i$ is not in the sample then $\mathbb{E}[(a^j - p^j) \cdot (q_i^j - p^j)] = \mathbb{E}[a^j - p^j] \cdot \mathbb{E}[q_i^j - p^j] = 0$. |
| | The bigger $\sum_{j=1}^k z_i^j$ is, the more we "suspect" element $i$ of being in the database. |

Table 1: Notation and intuition for Algorithm 2

only the query values for points in $X$) and, more subtly, because the analyst's decisions about how to sample the $p^j$'s, and which points in $X$ to include in the sets $A^j$, are independent of the query values outside of $X$. By the principle of deferred decisions, we may thus think of the query values $\{q_i^j : i \notin X, j \in [k]\}$ as selected after the interaction with the mechanism is complete.

Fix $i \notin X$. For every $j \in [k]$ and $i \notin X$, we have

$$\mathbb{E}\left[z_i^j\right] = \mathbb{E}\left[trunc_{3\varepsilon}(a^j - p^j) \cdot (q_i^j - p^j)\right] = \mathbb{E}\left[trunc_{3\varepsilon}(a^j - p^j)\right] \cdot \mathbb{E}\left[q_i^j - p^j\right] = 0.$$

By linearity of expectation, we also have $\mathbb{E}[z_i] = \mathbb{E}\left[\sum_{j=1}^k z_i^j\right] = 0$.

Next, note that $|z_i^j| \leq 3\varepsilon$, since $trunc_{3\varepsilon}(a^j - p^j) \in [-3\varepsilon, 3\varepsilon]$ and $q_i^j - p^j \in [0, 1]$. The terms $z_i^j$ are not independent, since if a partial sum $\sum_{j=1}^\ell z_i^j$ ever exceeds $\tau$, then subsequent values $z_i^j$ for $j > \ell$ will be set to 0. However, we may consider a related sequence given by sums of the terms $\tilde{z}_i^j = trunc_{3\varepsilon}(a^j - p^j) \cdot (\tilde{q}_i^j - p^j)$ (the difference from $z_i^j$ is that we use values $\tilde{q}_i^j$ $\mathsf{Ber}(p^j)$ regardless of whether item $i$ is in $A^j$). Once we have conditioned on the biases and mechanism's outputs, $\sum_{j=1}^k \tilde{z}_i$ is a sum of bounded independent random variables. By Hoeffding's Inequality, the sum is bounded $O(\varepsilon\sqrt{k \log(1/\varepsilon)})$ with high probability, for every $i \notin X$ $\mathbb{P}\left[\left|\sum_{j=1}^k \tilde{z}_i^j\right| > \varepsilon\sqrt{18k \ln\left(\frac{96}{\varepsilon}\right)}\right] \leq \frac{\varepsilon}{48}$.
By Etemadi's Inequality, a related bound holds uniformly over all the intermediate sums:

$$\forall i \notin X \quad \mathbb{P}\left[\exists \ell \in [k] \ : \ \left|\sum_{j=1}^\ell \tilde{z}_i^\ell\right| > \underbrace{3\varepsilon\sqrt{18k \ln\left(\frac{96}{\varepsilon}\right)}}_{\tau - 1}\right] \leq 3 \cdot \mathbb{P}\left[\left|\sum_{j=1}^k \tilde{z}_i^j\right| > \varepsilon\sqrt{18k \ln\left(\frac{96}{\varepsilon}\right)}\right] \leq \frac{\varepsilon}{16}$$

Finally, notice that by construction, the real scores $z_i^j$ are all set to 0 when an item is added to $A^j$, so the sets $A^j$ are nested ($A^j \subseteq A^{j+1}$), and a bound on partial sums of the $\tilde{z}_i^j$ applies equally well to the partial sums of the $z_i^j$. Thus, $\forall i \notin X$ $\mathbb{P}\left[\exists \ell \in [k] : \left|\sum_{j=1}^\ell z_i^\ell\right| > \tau - 1\right] \leq \frac{\varepsilon}{16}$

Now, the scores $z^i$ are independent across players (again, because we have fixed the biases $p^j$ and the mechanism's outputs). We can bound the probability that more than $\frac{\varepsilon N}{4}$ elements $i$ are "accused" over the course of the algorithm using Chernoff's bound: $\mathbb{P}\left[|A^k \setminus X| > \frac{\varepsilon}{8}N\right] \leq e^{-\varepsilon N/64} \leq e^{-\Omega(n)}$
The claim now follows by averaging over all of the choices we fixed. $\qquad\square$

The next claim states that the sum of the scores over all $i$ not in the sample is small.

**Claim 2.6.** *With probability at least* $\frac{99}{100}$, $\sum_{i \in [N] \setminus X} z_i = O(\varepsilon\sqrt{Nk})$.

*Proof.* Fix a choice of $(p^1, \ldots, p^k) \in [0, 1]^k$, the in-sample query values $(q_X^1, \ldots, q_X^k) \in \{0, 1\}^{n \times k}$, and the answers $(a^1, \ldots, a^k) \in [0, 1]^k$. Conditioned on these, the values $z_i$ for $i \notin X$ are independent and identically distributed. They have expectation 0 (see the proof of Claim 2.5) and are bounded by $\tau$ (by Claim 2.4). By Hoeffding's inequality, with probability at least $\frac{99}{100}$ $\sum_{i \in [N] \setminus X} z_i = O(\tau\sqrt{N}) = O(\varepsilon\sqrt{Nk})$ as desired. The claim now follows by averaging over all of the choices we fixed. $\qquad\square$

**Claim 2.7.** *There exists $c > 0$ such that, for all sufficiently small $\varepsilon$ and sufficiently large $n$, with probability at least $\frac{99}{100}$, either $\exists j \in [k] \ : \ |a^j - q^j(\mathcal{P})| > \varepsilon$ (large error), or $\sum_{i \in [N]} z_i \geq ck$ (high scores in sample).*

The proof of Claim 2.7 relies on the following key lemma. The lemma has appeared in various forms [20, 11, 21]; the form we use is [5, Lemma 3.6] (rescaled from $\{-1, +1\}$ to $\{0, 1\}$).

**Lemma 2.8** (Fingerprinting Lemma). *Let $f : \{0, 1\}^m \to [0, 1]$ be arbitrary. Sample $p \sim \mathsf{U}_{[0,1]}$ and sample $x_1, \ldots, x_m \sim \mathsf{Ber}(p)$ independently. Then*

$$\mathbb{E}\left[(f(x) - p) \cdot \sum_{i \in [m]} (x_i - p) + \left| f(x) - \frac{1}{m} \sum_{i \in [m]} x_i \right|\right] \geq \frac{1}{12}.$$

*Proof of Claim 2.7.* To make use of the fingerprinting lemma, we consider a variant of Algorithm 2 that does not truncate the quantity $a^j - p^j$ to the range $\pm 2\epsilon$ when computing the score $z_i^j$ for each element $i$. Specifically, we consider scores based on the quantities

$$\hat{z}_i^j = \begin{cases} (a^j - p^j) \cdot (q_i^j - p^j) & \text{if } i \notin A^j, \\ 0 & \text{if } i \in A^j; \end{cases} \qquad \text{and} \quad \hat{z}_i = \sum_{j=1}^{k} \hat{z}_i^j.$$

We prove two main statements: first, that these untruncated scores are equal to the truncated ones with high probability as long as the mechanism's answers are accurate. Second, that the expected sum of the untruncated scores is large. This gives us the desired final statement.

To relate the truncated and untruncated scores, consider the following three key events:

1. ("Few accusations"): Let $F$ the event that, at every round $j$, set of "accused" items outside of the sample is small: $|A_k \setminus X| \leq \varepsilon N/8$. Since the $A^j$ are nested, event $F$ implies the same condition for all $j$ in $[k]$.

2. ("Low population error"): Let $G$ be the event that at every round $j \in [k]$, the mechanism's anwer satisfies $|a^j - p^j| \leq 3\varepsilon$.

3. ("Representative queries"): Let $H$ be the event that $|\tilde{q}^j(\mathcal{P}) - p^j| \leq \varepsilon$ for all rounds $j \in [k]$— that is, each query's population average is close to the corresponding sampling bias $p^j$.

**Sub-Claim 2.9.** *Conditioned on $F \cap G \cap H$, the truncated and untruncated scores are equal. Specifically, $|a^j - p^j| \leq 3\varepsilon$ for all $j \in [k]$.*

*Proof.* We can bound the difference $|a^j - p^j|$ via the triangle inequality:

$$|a^j - p^j| \leq |a^j - q^j(\mathcal{P})| + |q^j(\mathcal{P}) - \tilde{q}^j(\mathcal{P})| + |\tilde{q}^j(\mathcal{P}) - p^j|.$$

The first term is the mechanism's sample error (bounded when $G$ occurs). The second is the distortion of the sample mean introduced by setting the query values of $i \in A^j$ to 0. This distortion is at most $|A_j|/N$. When $F$ occurs, $A^j$ has size at most $|X| + |A^j \setminus X| \leq n + \varepsilon N/8 = \varepsilon N/4$, so the second term is at most $\varepsilon/4$. Finally, the last term is bounded by $\varepsilon$ when $H$ occurs, by definition. The three terms add to at most $3\varepsilon$ when $F$, $G$, and $H$ all occur. $\square$

We can bound the probability of $H$ via a Chernoff bound: The probability of that a binomial random variable deviates from its mean by $\varepsilon N$ is at most $2\exp(-\varepsilon^2 N/3)$.

The technical core of the proof is the use of the fingerprinting lemma to analyze the difference $D$ between the sum of untruncated scores and the summed population errors: $D := \sum_{i=1}^{N} \hat{z}_i - \sum_{j=1}^{k} |a^j - q^j(\mathcal{P})| - k\mathbb{E}\left[\frac{|A^j|}{N - |A^j|}\right]$

**Sub-Claim 2.10.** $\mathbb{E}[D] = \Omega(k)$

*Proof.* We show that for each round $j$, the expected sum of scores for that round $\sum_i \hat{z}_i^j$ is at least $1/12 - \mathbb{E}\left[|a^j - q^j(\mathcal{P})| - \frac{|A^j|}{N - |A^j|}\right]$. This is true even when we condition on all the random choices and communication in rounds 1 through $j - 1$. Adding up these expectations over all rounds gives the desired expectation bound for $D$.

First, note that summing $z_i^j$ over all elements $i \in [N]$ is the same as summing over that round's unaccused elements $i \in [N] \setminus A^j$ (since $\tilde{z}_i^j = 0$ for $i \in A^j$). Thus,

$$\sum_{i=1}^{N} \tilde{z}_i^j = \sum_{i \in [N] \setminus A^j} \tilde{z}_i^j = (a^j - p^j) \sum_{i \in [N] \setminus A^j} (q_i^j - p^j).$$

We can now apply the Fingerprinting Lemma, with $m = N - |A^j|$, $p = p^j$, $x_i = \tilde{q}_i^j$ for $i \notin A^j$, and $f\left((x_i)_{i \notin A^j}\right) = a^j$ (note that $f$ depends implicitly on $A_j$, but since we condition on the outcome of previous rounds, we may take $A^j$ as fixed for round $j$). We obtain

$$\mathbb{E}\left[\sum_{i=1}^{N} \tilde{z}_i^j\right] \geq \frac{1}{12} - \mathbb{E}\left[\left|a^j - \frac{1}{N - |A^j|} \cdot \sum_{i \notin A^j} q_i^j\right|\right]$$

Now the difference between $\frac{1}{N-|A^j|} \sum_{i \notin A^j} q_i^j$ and the actual population mean $\frac{1}{N} \sum_{i=1}^{N} q_i^j$ is at most $N \cdot \left(\frac{1}{N} - \frac{1}{N-|A^j|}\right) = \frac{|A^j|}{N-|A^j|}$. Thus we can upper-bound the term inside the right-hand side expectation above by $|a^j - q^j(\mathcal{P})| + \frac{|A^j|}{N-|A^j|}$. □

A direct corollary of Sub-Claim 2.10 is that there is a constant $c' > 0$ such that, with probability at least $199/200$, $D \geq c'k$. Let's call that event $I$.

Conditioned on $F \cap G \cap H$, we know that each $\tilde{z}_i$ equals the real score $z_i$ (by the first sub-claim above), that $|a^j - q^j(\mathcal{P})| \leq 3\varepsilon$ for each $j$, and that $|A^k| \leq \varepsilon N/8$. If we also consider the intersection with $I$, then we have $D \geq c'k - 3k\varepsilon - k\frac{\varepsilon/8}{1-\varepsilon/8} \geq k(c' - 4\varepsilon)$ (for sufficiently small $\varepsilon$). By a union bound, the probability of $\neg(F \cap H \cap I)$ is at most $1/200 + \exp(-\Omega(\varepsilon^2 n)) \leq 1/100$ (for sufficiently large $n$). Thus we get $\mathbb{P}\left[(\neg G) \text{ or } \left(\sum_{i=1}^{N} z_i \geq ck\right)\right] \geq \frac{99}{100}$, where $c = c' - 4\varepsilon$ is positive for sufficiently small $\varepsilon$. This completes the proof of Claim 2.7. □

To complete the proof of the proposition, suppose that $|a^j - q^j(\mathcal{P})| \leq \varepsilon$ for every $j$, so that we can assume $\sum_{i \in X} z_i = \Omega(k)$. Then, we can show that, when $n$ is sufficiently large and $k \gtrsim \varepsilon^4 n^2$, the final query $q^*$ will violate robust generalization. A relatively straightforward calculation (omitted for space) shows that for the query $q^*$ that we defined, $q^*(X) - q^*(\mathcal{P}) = \Theta(\varepsilon\sqrt{k})$. Now, we choose an appropriate $k = \Theta(\varepsilon^4 n^2)$ we will have that $q^*(X) - q^*(\mathcal{P}) > \varepsilon$. By this choice of $k$, the first term in the final line above will be at least $2\varepsilon$. Also, we have $N \geq n = \Theta(\sqrt{k}/\varepsilon^2)$, so when $k$ is larger than some absolute constant, the $O(1/\sqrt{N})$ term in the final line above is $\Theta(\varepsilon/\sqrt[4]{k}) \leq \varepsilon$. Thus, by Claims 2.6 and 2.7, either $\mathcal{M}$ fails to be accurate, so that $\exists j \in [k]\ |a^j - q^j(\mathcal{P})| > \varepsilon$, or we find a query $q^*$ such that $q^*(X) - q^*(\mathcal{P}) > \varepsilon$.

### 2.3 Lower Bounds for All Algorithms via Random Masks

We prove Theorem 1.2 by constructing the following transformation from an adversary that defeats all natural algorithms to an adversary that defeats all algorithms. The main idea of the reduction is to use random masks to hide information about the evaluation of the queries at points outside of the dataset, which effectively forces the algorithm to behave like a natural algorithm because, intuitively, it does not know where to evaluate the query apart from on the dataset. The reduction is described in Algorithm 3. Due to space restrictions, we omit its analysis due to space.

## 3 Post Hoc Generalization Does Not Compose

In this section we prove that post hoc generalization is not closed under composition.

**Theorem 3.1.** *For every $n \in \mathbb{N}$ and every $\alpha > 0$ there is a collection of $\ell = O(\frac{1}{\alpha} \log n)$ algorithms $\mathcal{M}_1, \ldots, \mathcal{M}_\ell : (\{0,1\}^{5 \log n})^n \to \mathcal{Y}$ such that (1) for every $i = 1, \ldots, \ell$ and $\delta > 0$, $\mathcal{M}_i$ satisfies $(\varepsilon, \delta)$-post hoc generalization for $\varepsilon = O(\sqrt{\log(n/\delta)/n^{1-\alpha}})$, but (2) the composition $(\mathcal{M}_1, \ldots, \mathcal{M}_\ell)$ is not $\left(2 - \frac{2}{n^4}, 1 - \frac{1}{2n^3}\right)$-post hoc generalizing.*

The result is based on an algorithm that we call `Encrypermute`. Before proving Theorem 3.1, we introduce `Encrypermute` and establish the main property that it satisfies.

The key facts about `Encrypermute` are as follows.

---

**Algorithm 3:** $\mathcal{A}_{\mathsf{AQ}}$

---

Parameters: sample size $n$, universe size $N = \frac{8n}{\varepsilon}$, number of queries $k$, target accuracy $\varepsilon$.

Oracle: an adversary $\mathcal{A}_{\mathsf{NAQ}}$ for natural algorithms with sample size $n$, universe size $N$, number of queries $k$, target accuracy $\varepsilon$.

Let $\mathcal{X} = \{(i, y)\}_{i \in [N], y \in \{\pm 1\}^k}$

**For** $i \in [N]$
   Choose $m_i = (m_i^1, \ldots, m_i^k) \sim \mathsf{U}(\{\pm 1\}^k)$

Let $\mathcal{P}$ be the uniform distribution over pairs $(i, m_i)$ for $i \in [N]$

**For** $j \in [k]$
   Receive the query $\hat{q}^j : [N] \to [\pm 1]$ from $\mathcal{A}_{\mathsf{NAQ}}$
   Form the query $q^j(i, y) = y^j \oplus m_i^j \oplus \hat{q}^j(i)$ (NB: $q^j(i, m_i) = \hat{q}^j(i)$)
   Send the query $q^j$ to $\mathcal{M}$ and receive the answer $a^j$
   Send the answer $a^j$ to $\mathcal{A}_{\mathsf{NAQ}}$

---

---

**Algorithm 4:** `Encrypermute`

---

**Input:** Parameter $k$, and a sample $X = (x_1, x_2, \ldots, x_n) \in (\{0,1\}^d)^n$ for $d = 5 \log n$.

**If** *$X$ contains $n$ distinct elements*
   Let $\pi$ be the permutation that sorts $(x_1, \ldots, x_k)$ and identify $\pi$ with $r \in \{0, 1, \ldots, k! - 1\}$
   Let $\alpha \in [0, 1]$ be the largest number such that $k \geq n^\alpha$ and let $t \leftarrow \alpha k / 20$ (NB: $2^{dt} \leq k!$)
   Identify $(x_{k+1}, \ldots, x_{k+t}) \in (\{0,1\}^d)^t$ with a number $m \in \{0, 1, \ldots, k! - 1\}$
   **Return** $c = m + r \mod k!$

**Else**
   **Return** *a random number* $c \in \{0, 1, \ldots, k! - 1\}$

---

**Claim 3.2.** *Let $\mathcal{D}$ be any distribution over $(\{0,1\}^d)^n$. Let $D \sim \mathcal{D}$, let $X$ be a random permutation of $D$, and let $C \leftarrow \texttt{Encrypermute}(X)$. Then $D$ and $C$ are independent.*

Intuitively, the claim follows from the fact that $r$ is uniformly random and depends only on the permutation, so it is independent of $D$. Therefore $m + r \mod k!$ is random and independent of $m$.

**Lemma 3.3.** $\forall \delta > 0$, `Encrypermute` *satisfies $(\varepsilon, \delta)$-post hoc generalization for $\varepsilon = \sqrt{2 \ln(2/\delta)/n}$.*

Intuitively the lemma follows from the fact that $C$ is independent of $D$. We omit the proof of both of these claims due to space restrictions.

*Proof of Theorem 3.1.* Fix $\alpha \in (0, 1)$, and let $\mathcal{M}_1$ denote the mechanism that takes a database of size $n$ and outputs the first $n^\alpha$ elements of its sample. As $\mathcal{M}_1$ outputs a sublinear portion of its input, it satisfies post hoc generalization with strong parameters. Specifically, by [7, Lemma 3.5], $\mathcal{M}_1$ is $(\varepsilon, \delta)$-post hoc generalizing for $\varepsilon = O\left(\sqrt{\log(n/\delta)/n^{1-\alpha}}\right)$.

Now consider composing $\mathcal{M}_1$ with $O(\frac{1}{\alpha} \log n)$ copies of `Encrypermute`, with exponentially growing choices for the parameter $k$, where for the $i$th copy we set $k = (1 + \frac{\alpha}{20})^i \cdot n^\alpha$. By Lemma 3.3, each of these mechanisms satisfies post hoc generalization for $\varepsilon = O(\sqrt{\log(1/\delta)/n})$, so this composition satisfies the assumptions of the theorem.

Let $\mathcal{P}$ be the uniform distribution over $\{0,1\}^d$, where $d = 5 \log n$, and let $X \sim \mathcal{P}^{\otimes n}$. By a standard analysis, $X$ contains $n$ distinct elements with probability at least $\left(1 - \frac{1}{2n^3}\right)$. Assuming that this is the case, we have that the first copy of `Encrypermute` outputs $c = m + r \mod k!$, where $m$ encodes the rows of $X$ in positions $n^\alpha + 1, \ldots, (1 + \frac{\alpha}{20})n^\alpha$, and where $r$ is a deterministic function of the first $n^\alpha$ rows of $X$. Hence, when composed with $\mathcal{M}_1$, these two mechanism reveal the first $(1 + \frac{\alpha}{20})n^\alpha$ rows of $X$. By induction, the output of the composition of all the copies of `Encrypermute` with $\mathcal{M}_1$ reveals all of $X$. Hence, from the output this composition, we can define the predicate $q : \{0,1\}^d \to \{\pm 1\}$ that evaluates to 1 on every element of $X$, and to -1 otherwise. This predicate satisfies $q(X) = 1$ but $q(\mathcal{P}) \leq -1 + 2n/2^d = -1 + 2/n^4$. $\qquad \qquad \square$

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
