[Reviews · NeurIPS 2018]

Reviewer 1



This paper demonstrates some of the limits of mechanisms satisfying post hoc generalization, which guarantees accuracy in the adaptive data analysis setting. They show both an information-theoretic and computational lower bounds for sample complexity, and then show that mechanisms satisfying post hoc generalization do not in the worst case compose non-trivially. These results tie up some loose ends in adaptive data analysis, e.g. closing the previously-open gap in sample complexity for statistical queries, and gives a more complete picture of what is possible in this area. On the other hand, I would not call this the most clearly written paper. In particular, there is very little discussion or overview of the proofs. I recognize how difficult this is to do given the page limit constraints, but since a fair amount of the proofs are put into the appendix anyway, I consider it very important to communicate the ideas contained within the proofs rather than highlighting the technical details, as is done here. Nor does this paper highlight the difference between their strategy and what is contained in the previous literature. The basic strategy, after all, remains the same as in the lower bounds established in previous literature: Algorithm 2 attempts to uncover the sample given to M, then asks a query that M will overfit on, plus using fingerprinting, etc. What exactly is the new idea here? Also, to be blunt, why is the result that post hoc generalization does not compose interesting? After all, we know that we can achieve O(\sqrt{k}/\epsilon^2) sample complexity using differential privacy, and to achieve a better sample complexity, this paper shows that we can't guarantee post hoc generalization at all. It seems that when it comes to statistical queries, post hoc generalization is a bit besides the point, no? A few very small things: - Would be helpful to be much more explicit about sample complexity for uniform convergence, that way the reader can compare this to your results. (line 48) - Algorithm 2: A_1 should start empty: no one has been accused yet. - Algorithm 2: \tilde{q}_i^j presumably needs to lose the tilde. Edit (post author response): On the new idea in your results: I appreciate your attempt to stress the idea of an approximate attack rather than an exact attack. This seems like you're pointing out that a (very similar) attack, even if it doesn't work as well, still works well enough, which may or may not be of sufficient interest/novelty for your readers. On the interesting-ness of composition: AFAIK, there's no good way to directly show that an algorithm satisfies post hoc generalization without also satisfying DP, information bounds, or the like. This means it seems rather unlikely that you would actually need composition without already having stronger assumptions on the algorithms that you would like to compose, even if that assumption isn't DP. Which in turn means I don't know where this result would likely be relevant.

Reviewer 2



This paper studies a notion of generalization that allows multiple data-dependent interactions with the dataset. This notion is termed post-hoc generalization (or robust generalization in some previous work). The idea is that the output of the algorithm does not "reveal" any statistical query, i.e., the value of the query is similar on the dataset and the underlying population. Post-hoc generalization was studied in a COLT'16 paper by Cummings et al. More broadly, this paper is continuing the line of work on "adaptive data analysis" that has been studied in statistics for a while, and more recently in computer science (starting with a STOC'15 paper by Dwork et al.) The main contribution of the paper is a set of lower bounds for post-hoc generalization. In particular, the authors prove that: 1) A tight sample size lower bound of \sqrt{k}/eps^2 for any algorithm that can preserve k SQs up to error eps. Previously a lower bound of sqrt{k} was known for constant eps. 2) A similar computational lower bound (under cryptographic assumptions) that applies when the dimensionality of the data is smaller than the number of SQ queries. 3) In general, post-hoc generalization does NOT compose, in the worst-possible sense. (In contrast, it is known that special cases like differentially private algorithms are known to compose.) Overall, this is a nice paper that I believe should be accepted to NIPS. While I am not an expert in the techniques from the previous works (hence, I cannot judge the technical novelty given the time I spent on the paper), it seems to me that this is a solid contribution. At the conceptual level, the paper makes several interesting points that would be of value for the NIPS community.

Reviewer 3



Summary of paper contributions: This paper falls within the line of work on adaptive data analysis modeled on the basis of the statistical query model of Kearns. Their work is to show the limitations of algorithms that satisfy post hoc generalizations, as defined by Cummings et al. They show that 1. Algorithms satisfying post hoc generalization are require at least order k/\eps^2 samples to answer k adaptive queries with accuracy \eps, on datasets of dimension at least logarithmic in n. 2. They demonstrate a similar sample complexity lower bound for polynomial time algorithms, based on the existence of one-way functions. 3. Finally they show that algorithms satisfying post hoc generalizations do not compose: i.e. they construct O(\log n) algorithms satisfying nontrivial post hoc generalization whose composition has essentially trivial post hoc generalization. Summary opinion: The paper is well-written and has interesting lower bound results on the post hoc generalization notion of Cummings et al. It will be of significant interest to the community and a good addition to the program. Consequently, I recommend its acceptance. Comments for authors: 1. For the natural algorithm lower bound, it is not mentioned what the sample X \subset [N] is. Presumably the definition is something simple. 2. Similarly there seems to be some notational inconsistency e.g. q^j_i vs q^j(i) for algorithm 2. 3. It is not clear in some places whether the notation O() or \Theta requires to be used, e.g. in Theorem 3.1. It improves clarity for the reader to be careful with this. I have read the author response. I would strongly recommend to the authors to include a proof overview highlighting and expanding the differences and technical novelty of their techniques to prior work, as they indicated in the response.